# Multiple memories can be simultaneously reactivated during sleep as effectively as a single memory

Eitan Schechtman [1✉], James W. Antony[2], Anna Lampe[1], Brianna J. Wilson[1], Kenneth A. Norman [2] & Ken A. Paller [1]

Memory consolidation involves the reactivation of memory traces during sleep. If different memories are reactivated each night, how much do they interfere with one another? We examined whether reactivating multiple memories incurs a cost to sleep-related benefits by contrasting reactivation of multiple memories versus single memories during sleep. First, participants learned the on-screen location of different objects. Each object was part of a semantically coherent group comprised of either one, two, or six items (e.g., six different cats). During sleep, sounds were unobtrusively presented to reactivate memories for half of the groups (e.g., "meow"). Memory benefits for cued versus non-cued items were independent of the number of items in the group, suggesting that reactivation occurs in a simultaneous and promiscuous manner. Intriguingly, sleep spindles and delta-theta power modulations were sensitive to group size, reflecting the extent of previous learning. Our results demonstrate that multiple memories may be consolidated in parallel without compromising each memory's sleep-related benefit. These findings highlight alternative models for parallel consolidation that should be considered in future studies.

[1] Department of Psychology, Northwestern University, Evanston, IL 60208, USA. [2] Princeton Neuroscience Institute and Department of Psychology, Princeton University, Princeton, NJ 08544, USA. ✉email: eitan.schechtman@northwestern.edu

More than a century after researchers started to explore the beneficial effects of sleep on memory[1], the mechanism by which this benefit is achieved is still debated[2]. A leading hypothesis, termed active systems consolidation[3], postulates that memories stored in the hippocampus are reactivated during sleep, subsequently shaping neocortical memory traces. Reactivation of memories during sleep was first observed in rodents[4,5]. Sequential learning-related spiking activity was shown to "replay" during sleep, and this phenomenon has since been connected to the offline consolidation process[6]. In humans, recent studies using multivariate pattern classification with fMRI and magnetoencephalography (MEG) have shown evidence for reactivation of cortical and hippocampal memory-related patterns during sleep and awake rest[7–12].

These findings in humans and non-human animals have been instrumental in revealing how individual memories are consolidated during sleep. Over a typical night of sleep, a multitude of such memories are presumably consolidated, each reactivated to improve future retention. The question of how the brain enables many independent reactivation processes to occur and how the resources involved in the process are managed to avoid interference is relatively unexplored[13,14]. Although resource availability may be sufficient to allow for all relevant memories to undergo reactivation throughout the course of a night, we sought to consider whether the brain can support multiple reactivation processes occurring at once. Crucially, we wanted to test whether parallel reactivation incurs a toll on memory benefits, implying that simultaneous processing exhausts the resources critical for memory consolidation.

One line of evidence that seems to support the notion of parallel reactivation comes from studies using targeted memory reactivation (TMR). TMR involves the unobtrusive presentation of learning-related (commonly auditory or olfactory) stimuli during sleep to selectively reactivate specific memories[15]. Whereas olfactory designs have reactivated multiple learned items using a single odor (e.g., 15 items[16–18]), auditory designs have commonly used sounds that were associated with a single item (and more recently, with two items[19,20]). A recent meta-analysis has shown that both techniques consistently benefit cued items[21], but no prior study has directly compared the memory benefits of reactivating a single item versus multiple items.

The results from olfactory TMR studies demonstrate reactivation of multiple memories using a single odor cue, but are the memories reactivated simultaneously? Simultaneous reactivation is but one of several ways to explain the observed benefit. Another possibility, for example, is that a single memory was sampled and reactivated with each odor cue presentation (or with each sniff during odor presentation blocks). The overall results would still demonstrate a benefit of cuing over all items, but each item should gain less than it would have if all items were reactivated independently.

In order to more fully test the possibility of simultaneous reactivation, it is necessary to consider whether memory benefits rely on the number of items reactivated. The relationship between the number of items cued and the cuing benefit may shed light on the general attributes of sleep reactivation, revealing whether memories are indeed reactivated simultaneously or whether limited reactivation resources strictly govern sleep-related consolidation and limit it to a small number of items at any given time.

To examine this question, we contrasted the effects of TMR for single items and larger groups of items within the same study design (i.e., within subjects) and using the same cuing modality. In a spatial-memory task with auditory TMR, we required participants to learn the locations of 90 images on a 2D circular grid (Fig. 1a). Items were strategically grouped into distinct sets, and

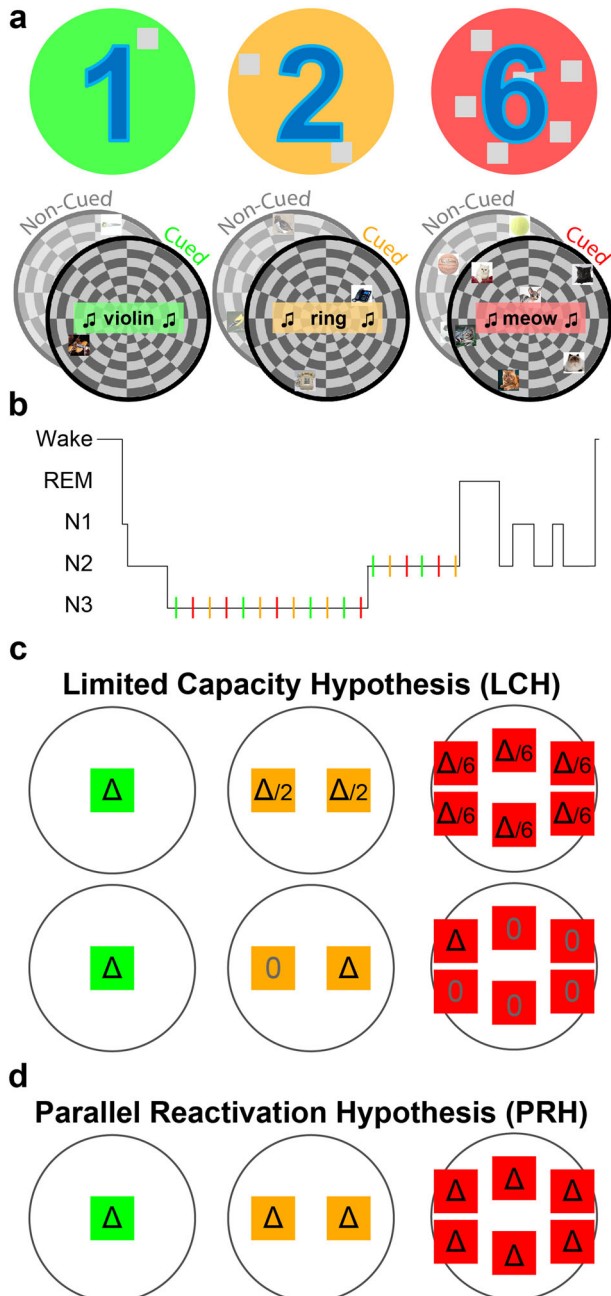

Fig. 1 Design and predictions. a In a spatial-memory task, participants learned the specific locations of images that appeared on a circular grid. Images belonged to sets that included six images, two images, or one image (e.g., the cat set consisted of six distinct images of different cats whereas the phone set consisted of two phones). Each set was associated with a set-specific sound (e.g., "meow", "ring"). Some sets were cued during subsequent sleep (colored sets) and others were not (gray sets). b Sounds were presented during stages N2 and N3 in an afternoon nap. c The limited capacity hypothesis (LCH) predicts that each item belonging to a large set would benefit less from TMR relative to items belonging to a smaller set. If a single-item set (green) receives a certain benefit (Δ), this benefit will be divided between items in the two-item set (orange) and the six-item set (green). This benefit could, for example, be split evenly between items (top) or designated to a single item of the set (bottom). Other variants of this hypothesis are explored in the main text. d The parallel reactivation hypothesis (PRH; center) predicts that the benefit per item would not depend on group size. Colors, symbols same as c.

**Table 1 Sleep architecture (mean ± SEM).**

|  | Total time with lights off | Wake | N1 | N2 | N3 | REM |
|---|---|---|---|---|---|---|
| Minutes | 93.23 ± 0.59 | 18.68 ± 2.72 | 10.15 ± 1.03 | 22.90 ± 2.26 | 36.06 ± 2.76 | 5.44 ± 1.59 |
| Percentage | 100% | 20.06 ± 2.94 | 10.81 ± 1.05 | 24.59 ± 2.44 | 38.71 ± 3.00 | 5.83 ± 1.67 |

each set included either a single item, two items, or six items (e.g., six distinguishable images of different cats). Each set was associated with a single unique sound (e.g., a single meow linked to all six cats). Importantly, each set of related items shared a semantic relation with other items in their group, but the spatial position of each item was unique and not associated with that of other items in the group. The locations of unique items within a set were learned to criterion in separate blocks to minimize competition within a group and avoid within-set interactions that may facilitate memory for item locations. Participants were next allowed a 90-min nap opportunity. During non-REM sleep (NREM), half of the sounds for each set size were unobtrusively presented (Fig. 1b). The benefits of TMR were compared between cued and non-cued items.

Using this design, we aimed to tease apart two main hypotheses regarding the neurocognitive mechanisms by which memory reactivation achieves its benefit during sleep, the limited capacity hypothesis (LCH; Fig. 1c) and the parallel reactivation hypothesis (PRH; Fig. 1d):

First, if reactivating multiple items involves the separate reactivation of each item, in a manner similar to path-specific replay, these activations may each incur some cost, or may rely on common limited resources. Previous studies have suggested that the capacity of memory reactivation is limited in the sense that increasing reactivation for some memories during a period of sleep may decrease the likelihood that others will be reactivated, theoretically due to a limited time window for consolidation during sleep[22–24]. An underlying assumption behind this hypothesis is that reactivation cannot occur in parallel, simultaneously, for multiple items. If the reactivation of two memories could happen independently in parallel, they would not need to compete one with the other for reactivation time. When considering multiple memories cued at the same time, the Limited Capacity Hypothesis (LCH; Fig. 1c) therefore assumes a finite capacity for simultaneous reactivation during sleep. By this view, if multiple items are reactivated together, the benefit of reactivation is split among all items associated with the presented cue: it could be divided between items so that each gets a share of the benefit (as shown in Fig. 1c); or it could be allotted so that only a subset of items benefit whereas the others do not. A recent study provided evidence supporting this latter option; Antony et al.[19] found that only one of a pair of simultaneously cued items but not the other benefited, suggesting that sleep-related reactivation may be limited to a single item at any given time. If only a subset of items is selected for reactivation, they may be selected randomly with each presentation of a cue or there could be some systematic bias towards reactivating a specific set of items (e.g., weakly learned memories might be more likely reactivated[8,25]).

Alternatively, reactivation of multiple memories may be achieved in a parallel, independent manner. The parallel reactivation hypothesis (PRH; Fig. 1d) suggests that there is no cost to simultaneously reactivating multiple items at the same time as opposed to a single item. To our knowledge, parallel reactivation during sleep has never been argued for or directly tested, but it is in line with reports of similar effect sizes found in multi-item and single-item TMR studies as assessed by meta-analysis[21]. In our design, this hypothesis predicts similar TMR benefits to all items in a set, regardless of set size.

Although these hypotheses are not mutually exclusive, our design contrasted these hypotheses in a controlled setting. The results challenge the LCH, suggesting that reactivation may be simultaneous and, in a sense, not resource-limited. We use these findings to question whether sleep-related consolidation is necessarily based on reactivation of individual memories as opposed to a wider, more generalized context or semantic structure.

## Results

**The benefit of sleep reactivation was uninfluenced by set size.** Training consisted of feedback-based trials during which participants learned the spatial locations of 90 images to a 100-pixel criterion. On average, participants needed $3.63 \pm 0.15$ trials (mean ± SEM) to reach criterion per image. The number of trials was not significantly different for cued and non-cued sets ($F(1, 30) = 0.66$, $p = 0.42$, $\eta^2 = 0.02$) or for different set sizes ($F(2, 60) = 1.21$, $p = 0.30$, $\eta^2 = 0.04$), nor was there a significant interaction between the two factors ($F(2, 60) = 1.58$, $p = 0.21$, $\eta^2 = 0.05$). During the NREM stages of the subsequent nap (Table 1), sounds associated with different sets were unobtrusively presented ($11 \pm 1$ repetitions per sound, of which $10.19 \pm 1.1$ were during sleep stage N3). The number of repetitions per sound was not significantly different for sounds associated with different set sizes ($F(2, 60) = 1.07$, $p = 0.35$, $\eta^2 = 0.03$), nor was there an interaction between set size and stage of NREM sleep (N2 vs N3; $F(2, 60) = 0.3$, $p = 0.74$, $\eta^2 = 0.01$).

Spatial-memory for learned items was evaluated before and after sleep. In previous spatial-memory TMR paradigms with single distinct images, each associated with a unique sound, spatial accuracy was measured as the pixel distance between correct and recalled positions[20,26]. Our task included a categorical structure, whereby several distinguishable images (e.g., different cats) were associated with the same sound (e.g., meow). Consequently, we could ascertain the influence of set size on the benefit of TMR. However, another source of error that may vary with set size is that participants may mistakenly recall the location of an object other than the one they attempted to recall. For example, participants may suffer a memory confusion among the different cats and mistakenly place one cat (the Persian cat) in the location of another from the same set (the Siamese cat). We therefore designed the study in a manner that allowed us to dissociate these errors, which we term swap errors, from placement errors that more directly reflect degree of spatial accuracy in recall (see Methods section).

Focusing first only on sets with two or six items, we considered the effects of set size and cuing during sleep on swap errors. The average number of swap errors for an item in a six-item set was $0.385 \pm 0.02$ before sleep and $0.391 \pm 0.02$ after sleep (change = $+0.005$). The average number of swap errors for an item in a two-item set was $0.175 \pm 0.02$ before sleep and $0.178 \pm 0.02$ after sleep (change = $+0.003$). Comparing the effects of set size and future-cuing status on pre-sleep swap errors using a repeated-measures ANOVA revealed a significant effect of set size, with larger sets having more swap errors ($F(1, 30) = 58.9$, $p < 0.001$, $\eta^2 = 0.66$). Importantly, there were no pre-sleep differences between cued and non-cued sets ($F(1, 30) = 0.006$, $p = 0.94$, $\eta^2 = 0.0002$), nor was there a significant interaction ($F(1, 30) = 0.08$, $p = 0.78$,

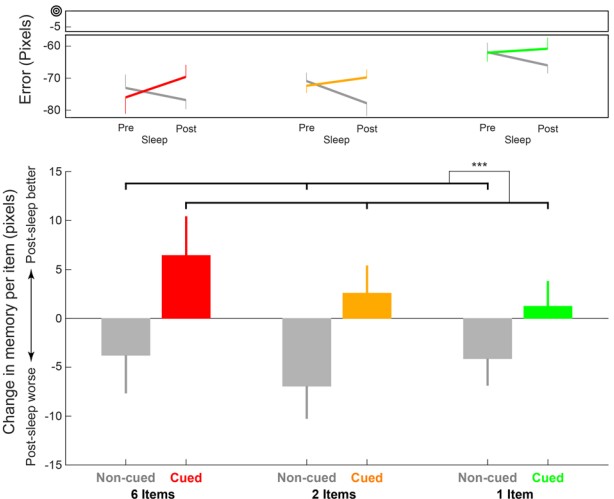

**Fig. 2 Cuing status, but not set size, affected sleep-related benefits.** The upper panel shows mean error rates before and after sleep as a function of set size (red – six-item sets; yellow – two-item sets; and green – one-item sets) and cuing status (colored – cued; gray – non-cued). Zero error signifies exact recall of the correct location. The lower panel shows the change in memory over sleep (i.e., sleep-related benefit). Negative values represent higher errors after sleep. There was a significant effect of cuing status but no significant effect of set size and no cuing by set size interaction. Error bars signify standard error of the mean (between-subjects). ***$p < 0.001$.

$\eta^2 = 0.003$). We next turned to changes to memory between pre- and post-sleep tests. Using a repeated-measures ANOVA, we found no main effect of cuing on the change in swap errors ($F$ (1, 30) = 0.43, $p = 0.52$, $\eta^2 = 0.014$), no main effect of set size ($F$ (1, 30) = 0.03, $p = 0.87$, $\eta^2 = 0.0009$), and no interaction between cuing and set size ($F$ (1, 30) = 0.24, $p = 0.63$, $\eta^2 = 0.008$).

Previous spatial TMR studies have found effects of cuing on accuracy[20,25,26]. For all subsequent analyses, we therefore focused on accuracy errors and included only items that were not considered as swapped (see Methods section). We first analyzed pre-sleep test results to confirm that they did not differ between cued and non-cued sets. Using a repeated-measures ANOVA we found that pre-sleep error rates were not different between cued and non-cued sets ($F$ (1, 30) = 0.74, $p = 0.4$, $\eta^2 = 0.024$), nor was there an interaction between cuing status and set size ($F$ (2, 60) = 0.14. $p = 0.87$, $\eta^2 = 0.005$). Error rates differed between set sizes ($F$ (2, 60) = 7.52, $p < 0.01$, $\eta^2 = 0.2$), with greater errors for sets of six and two items relative to single items ($p < 0.01$, Tukey's HSD), but no significant difference between two- and six-item sets ($p > 0.7$, Tukey's HSD; Fig. 2).

The two aforementioned hypotheses make different predictions regarding how set size would influence cuing. To test these effects, we calculated, per participant, the average change in spatial error over the period of sleep for different set sizes (six, two, or one) and cuing status (cued or non-cued). As shown in Fig. 2, cuing influenced recall accuracy ($F$ (1, 30) = 13.4, $p < 0.001$, $\eta^2 = 0.31$), producing superior recall for cued compared to non-cued locations (see Supplementary Fig. 1 for individual participant results). However, accuracy was not modulated by set size ($F$ (2, 60) = 0.66, $p = 0.52$, $\eta^2 = 0.02$) or the interaction between set size and cuing status ($F$ (2, 60) = 0.42, $p = 0.66$, $\eta^2 = 0.014$). Using the Bayesian Information Criterion to estimate the Bayes factor[27–29] indicated strong evidence that there was no interaction effect ($BF_{01} \approx 25$, equal priors for $H_0$ and $H_1$). On average, spatial-memory error after sleep decreased by $2.98 \pm 1.87$ pixels for cued sets and increased by $4.76 \pm 2.39$ pixels for non-cued sets.

These results pertaining to set size, however, could potentially have been influenced by the aforementioned differences in pre-sleep accuracy among sets of different sizes (i.e., error rates were significantly smaller for one-item sets). To deal with this possible bias, we conducted two complementary analyses (Supplementary Fig. 2). First, we regressed out pre-sleep results to eliminate set-dependent effects of regression to the mean[19,25,30] (see Methods section). This analysis produced similar results, with a significant effect for cuing ($F$ (1, 30) = 14.75, $p < 0.001$, $\eta^2 = 0.33$), no significant effect of set size ($F$ (1, 30) = 2.16, $p = 0.12$, $\eta^2 = 0.07$), and no significant interaction between set size and cuing status ($F$ (2, 60) = 0.29, $p = 0.75$, $\eta^2 = 0.01$; Supplementary Fig. 2a). For the second analysis, we chose 500 subsampled versions of our dataset that eliminated significant pre-sleep differences related to set size (see Methods section). The results were consistent with those obtained for the complete dataset, with 67% of the subsampled datasets showing a significant ($p < 0.05$) cuing effect (mean($p$) = 0.061, median($p$) = 0.024) and only 0.6% showing a significant ($p < 0.05$) cuing by set size interaction (mean($p$) = 0.6, median($p$) = 0.62; Supplementary Fig. 2b). All 500 subsampled datasets showed higher sleep-related benefits for the cued relative to the non-cued sets. As an additional measure, we calculated the skewness of both $p$-value distributions and found a strong left-leaning tendency for the cuing effect $p$-value distribution (skewness = 3.05), indicating the peak of the distribution is at the lower $p$-value levels. Conversely, the interaction $p$-value distribution showed a moderate right-leaning tendency for the interaction effect $p$-value distribution (skewness = -0.26), indicating that the histogram does not show any clear peak for lower $p$-values.

An inherent difference between six-, two-, and single-item sets is that sounds associated with larger sets were presented more frequently during training (i.e., because they were coupled with more individual images). The number of times participants were presented with each sound was therefore highly dependent on set size. To consider the effect familiarity may have on our results, we used the variability in the number of trials (and sound repetitions) between sets of the same size (e.g., some six-item sounds were presented more than others). We reanalyzed our data over all sets, using this familiarity index as a covariate, and found qualitatively similar results ($p < 0.02$ for the cuing effect; $p = 0.74$ for the interaction effect), indicating that sound familiarity did not substantially contribute to the observed effect.

Because training involved a binary criterion short of perfection, in that placements within 100 pixels from the target were deemed correct, participants may have been shaped to ignore small errors in accuracy. We therefore repeated the analysis including a lenient measure of accuracy – the number of items considered correct using the training criterion. There were 75.91% of items considered correct before sleep across conditions and participants. The effects of TMR were qualitatively similar, with an effect of cuing ($F$ (1, 30) = 17.33, $p < 0.001$, $\eta^2 = 0.37$), no effect of set size ($F$ (2, 60) = 0.4, $p = 0.67$, $\eta^2 = 0.01$), and no interaction ($F$ (2, 60) = 0.43, $p = 0.65$, $\eta^2 = 0.01$). On average, the number of correct trials increased by 0.04 for cued sets and decreased by 0.04 for non-cued sets.

Our results show that the cuing-related benefit of a single item does not depend on the size of the set to which it belongs. To complement this analysis, we considered the cumulative benefit for sets of different sizes by computing the sum of all benefits aggregated over the different items within a set. Like the per-item analysis, the per-set analysis also revealed a significant effect of cuing ($F$ (1, 30) = 9.83, $p < 0.01$, $\eta^2 = 0.25$) and no main effect of set size ($F$ (2, 60) = 0.35, $p = 0.71$, $\eta^2 = 0.01$). However, unlike the per-item analysis, this analysis revealed a marginal interaction ($F$ (2, 60) = 3.02, $p = 0.056$, $\eta^2 = 0.09$), indicating a trend for

larger cumulative benefits for larger sets relative to smaller ones. Given similar benefits for single items within a set, at the level of whole sets it would be reasonable for a set with many items to benefit more than a set with few items.

In conclusion, the results presented in this section indicate that set size within the range sampled did not influence the benefit of sleep for the individual items within a set. The notion that there is a finite capacity for reactivation during sleep (the LCH), favoring the smaller set size, is therefore not supported by the results.

**Within-set relationships did not support a model whereby only a subset of items was reactivated**. The previous results challenge the LCH (Fig. 1c), regardless of whether benefits from a limited capacity for reactivation are split among all items or focused more on a subset of items. However, our previous results[19] supporting the LCH model motivated us to further examine these issues[19]. The predictions made by this hypothesis depend on whether an item (or a small subset of items) is randomly selected and reactivated with each cue presentation during sleep (i.e., a random subset reactivation model) or whether some bias governs which item will be selected and reactivated over multiple presentations (i.e., a biased subset reactivation model). To examine the latter, we calculated the intra-class correlation (ICC) of the sleep-related change in accuracy error for sets consisting of six and two items[31,32]. The ICC, commonly used in reliability analyses, provides a measure of total agreement among measurements within a set. If a small subset of items within a set benefited from cuing whereas the rest were on par with items of non-cued sets, one would expect higher agreement for non-cued than for cued sets. However, our results showed that the ICC was not different between cued and non-cued sets, regardless of set size ($p = 0.45$ for six items; $p = 0.81$ for two items; Fig. 3a).

Although ICC is essentially a measure of variability, large variability does not necessarily stem from a subset of deviant measurements (i.e., it could be due to a wider distribution). To complement the ICC analysis, we considered a primitive measure of deviance within set. For each six-item set, we calculated the Z-scores of the sleep-related change in error for each item and used the maximum absolute value as a representative statistic to consider whether the set includes an outlier-like data point. A similar analysis was not conducted for two-item sets because outliers are not well defined within a pair. These scores were not different between cued and non-cued sets of six items ($t (119) = -1.1$, $p = 0.27$, Cohen's $d = 0.2$; Fig. 3b). Taken together, the ICC and Z-score analyses did not support the biased subset reactivation model.

An alternative model posits that a small, random subset of items would be reactivated with each cue presentation during sleep. This hypothesis is inconsistent with the lack of an effect of set size on cuing benefit, as described above. If items were randomly sampled with each cue presentation, the probability for an item within a larger set to benefit from the cue would be smaller than the probability for an item within a smaller set. Like the divided benefit model, this model would also predict a graded, set-size-dependent benefit of cuing on memory with larger benefits for smaller sets. This prediction was not supported by our data.

Another prediction, based on the random subset hypothesis, is that memory benefit would correlate with the number of times a cue was presented during sleep. Previous studies with single items have often failed to show correlations between the number of repetitions of a cue during sleep and the benefit of TMR[33,34]. However, if random sampling occurred and only one item was reactivated with each cue presentation, the probability of an item

being reactivated (in a set with more than a single item) should be proportional to the number of cue repetitions (e.g., the probability for an item in a six-item set to be reactivated at least once is $1 - \left(\frac{5}{6}\right)^n$ with $n$ being the number of repetitions: 0.17, 0.31, and 0.42 for one, two, or three repetitions, respectively). Therefore, the random subset reactivation model would predict a positive correlation between the average cuing benefit for a multi-item set (i.e., two- and six-item sets) and the number of repetitions of the associated sounds over sleep.

The mean ($\pm$SEM) number of repetitions for the six-, two-, and one-item sets were 11.05 ($\pm 1$), 11.04 ($\pm 1$), and 10.98 ($\pm 1$), respectively. Consistent with previous studies, there was no significant correlation between the number of repetitions of sounds for single-item sets and the benefit of cuing ($r = 0.09$, $p = 0.63$; Fig. 3c). Importantly, correlations for multi-item sets were also nonsignificant ($r = 0.01$, $p = 0.95$ for sets of two items; $r = 0.29$, $p = 0.29$ for sets of six items; Fig. 3c). We also tested whether cuing benefits correlated with the probability of an item being reactivated, as described above, and found no significant correlation ($r = 0.19$, $p = 0.31$ for sets of two items; $r = 0.13$, $p = 0.48$ for sets of six items).

Taken together, these data contradict the predictions made by the random subset reactivation model. Altogether, our results do not support the idea that a subset of items benefits from TMR, making PRH the only hypothesis that is not challenged by our findings. Despite the fact that this conclusion turns on null findings, we believe that our analytic approach of addressing the different hypotheses using multiple complementary analyses provides reasonable evidence for this claim.

**Delta-theta and sigma activity after cue presentation during sleep was modulated by set size**. Throughout the experiment, we collected scalp electroencephalographic data. We used these data for sleep staging and also to analyze responses to cue presentation during sleep, focusing on two frequency bands shown to be modulated by cues in previous studies[30,35]. The sigma range (11–16 Hz) encompasses sleep spindles, which are ramped waves lasting between 500 and 3000 ms. Spindles have been linked to memory consolidation (see Antony et al.[36] for review). Consolidation has also been linked with rhythms in the delta and theta bands (0.5–4 Hz and 4–8 Hz. respectively). Slow waves are the defining characteristic of N3, the deepest stage of NREM sleep, and consist mostly of activity in the delta band. K-complexes are similar waveforms consisting of a fast negative peak and a slower positive component that appear individually during stage N2 and N3 and involve component in both the theta and delta frequency ranges. To investigate cue-related modulations during sleep, we averaged the mean time-frequency response profile across participants (Fig. 4a, top). We identified two clusters in time and frequency that were significantly modulated by sounds ($p < 0.001$, corrected; Fig. 4a, bottom). These clusters corresponded roughly with the delta-theta band (0–11 Hz, 0.31–0.87 s after cue onset) and the sigma band (11.5–17.25 Hz, 0.94–1.44 s), as reported in other studies examining cue-related modulations during sleep[35].

Considering these two clusters, we examined the power modulation for each set size (Fig. 4b). In addition to playing the sounds that had been associated with one, two, or six items before sleep, we presented a single novel sound as well and included data from that condition in these analyses ($n = 2823$, 2814, 936, and 318 trials across participants for the one-, two-, six-item sounds and the novel sound, respectively). By using a linear mixed model, we were able to account for differences in trial counts between set sizes and participants (as reflected in Fig. 3c), while harnessing the variability within participants and

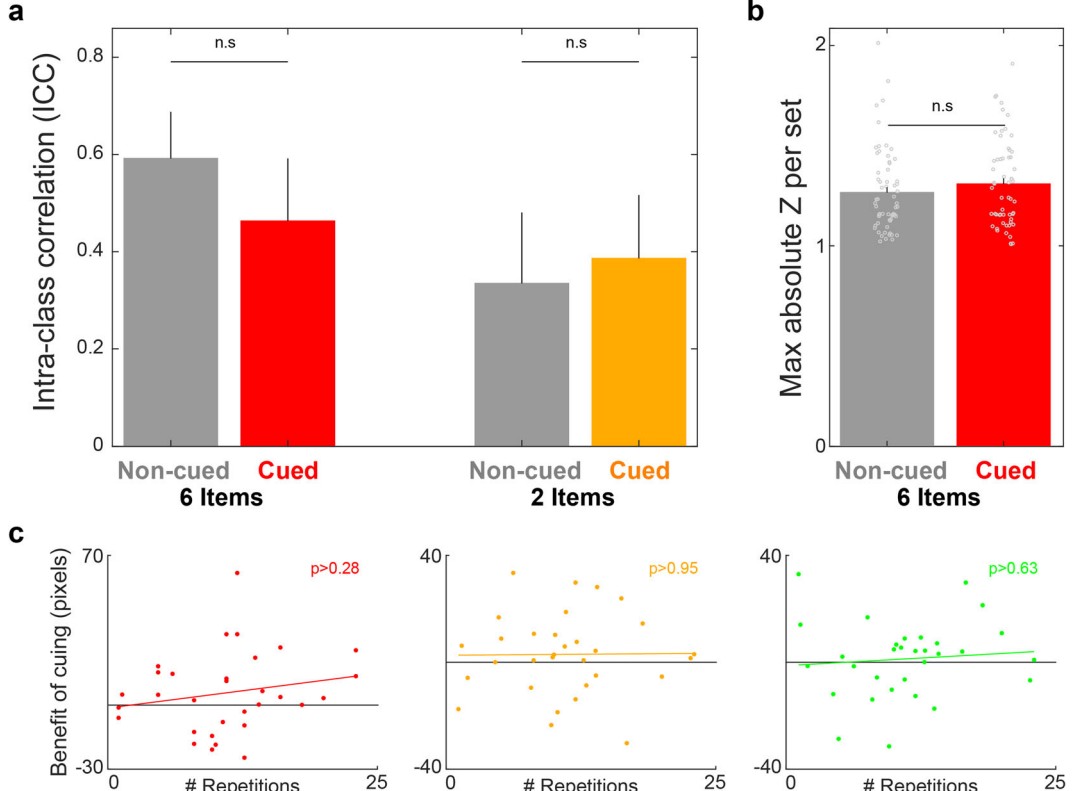

**Fig. 3 Models predicting that a subset of items benefits from reactivation are not supported by the behavioral data. a** If a small subset of items were to benefit from cuing in a biased manner (i.e., biased subset reactivation model), we would expect significantly higher interclass correlations between benefits of individual items for non-cued sets relative to cued sets, but no such effect was observed. **b** Another prediction of the biased subset reactivation model is that the cued sets would be more prone to outlier-like results (e.g., a single item's benefit would be much higher than the others'). Comparisons between the maximal absolute $Z$-score within set for cued and non-cued sets of six items failed to show a significant difference on this measure. **c** If a small subset of items randomly benefited from each presentation (i.e., random subset reactivation model), we would expect that more repetitions per cue would result in larger benefits of cuing to memory (relative to non-cued items), but these correlations were not significant for any set size (red – six-item sets; yellow – two-item sets; and green – one-item sets). Error bars signify standard error of the mean. n.s. nonsignificant comparisons.

sounds to our advantage (see Methods section). Results showed that power for both the delta-theta and the sigma clusters was linearly modulated by the number of items previously associated with the sound (delta-theta: $t$ (6889) = 2.52, $p < 0.02$, modulation coefficient = 1.97, 95% CI: 0.44 to 3.51; sigma: $t$ (6889) = 2.08, $p < 0.04$, modulation coefficient = 1.99, 95% CI: 0.11 to 3.87; Fig. 4c). The estimated coefficients imply that larger set sizes were linearly correlated with larger modulations in both identified frequency bands.

This measure of power in the sigma range may be affected by two factors: spindle probability and spindle amplitude. Therefore, the sigma results could represent three scenarios: (a) more spindles followed sounds associated with more items; (b) a similar number of spindles occurred, but with lower amplitudes for smaller set sizes; or (c) both spindle probability and amplitude were modulated by the number of associated items. To disentangle these scenarios, we used an automated algorithm to detect spindles in single trials and calculated the probability of a spindle taking place at any given time point following cue onset. The change in spindle probability, calculated by subtracting spindle probability during baseline (see Methods section) was compared for sounds associated with different set sizes (Fig. 4d). Considering the time frame identified for the previously described cluster, we found that spindle probability was linearly and positively modulated by the number of associated items ($t$ (6889) = 2.22, $p < 0.03$, modulation coefficient = 0.01, 95% CI:

0.001 to 0.02). In contrast, the maximum amplitude for all spindles that followed a cue was not significantly modulated by set size ($t$ (1608) = −0.44, $p = 0.65$, modulation coefficient = −0.04, 95% CI: −0.22 to 0.14; Fig. 4e). Therefore, our data support the first outlined scenario, whereby more spindles follow sounds associated with more items.

As mentioned with respect to the behavioral results, the familiarity of different sounds due to differences in the number of times they were presented during training may have confounded the physiological results. If so, delta-theta power, sigma power, and spindle probability would be modulated by the number of repetitions for each sound – but this was not the case ($p = 0.25$, modulation coefficient = −0.92 for delta-theta power; $p = 0.43$, modulation coefficient = −0.71 for sigma power; $p = 0.7$, modulation coefficient = −0.002 for spindle probability).

Finally, we also tested whether delta-theta power, sigma power, or spindle probability modulated memory performance for individual items in sets of different sizes. Using a linear mixed model to predict cuing benefit per item, we found no significant effect of any of the three physiological measures ($p = 0.56$, $p = 0.9$, $p = 0.64$, for delta-theta power, sigma power, and spindle probability, respectively), nor did we find a significant coefficient for the interaction component ($p = 0.81$, $p = 0.37$, $p = 0.83$, for delta-theta power, sigma power, and spindle probability, respectively). The cuing effect was also not correlated with total sleep time ($p = 0.28$) or total time in NREM sleep ($p = 0.35$).

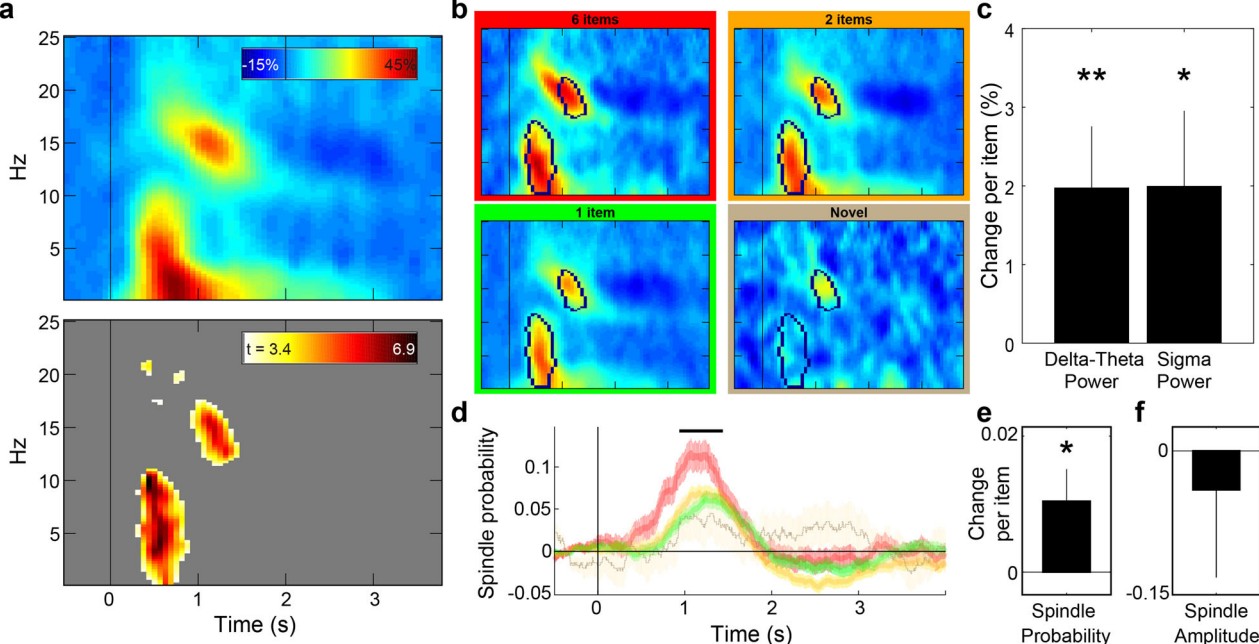

**Fig. 4 Sleep spindles and delta-theta power following cue presentation were influenced by set size. a** Top - spectrograms displaying the baseline-corrected time-frequency responses to cue presentations (regardless of set size) during sleep averaged over participants. Time zero represents cue onset during sleep. Bottom - t-value map reflecting threshold-crossing (p < 0.001, corrected) points in the time-frequency space. Two major clusters reflect delta-theta (0–11 Hz) and sigma (11.5–17.5 Hz) evoked activity following cue onset. **b** Spectrograms averaged across trials for the different set sizes. Black outlines mark the identified clusters. **c** Modulation coefficient of the number of items associated with a sound on delta-theta (left) and sigma (right) power (i.e., the change in power with each additional item). **d** Time-locked modulation of spindle probability following cue onset for different set sizes. Black line marks the timeframe detected for the sigma cluster, which was used for the analyses shown in the next panels. **e** Modulation coefficient of the number of items associated with a sound on spindle probability. **f** Modulation coefficient of the number of items associated with a sound on the amplitude of spindles occurring after sound onset. Error bars and shaded areas signify standard error of the mean. Data from electrode location Cz. **p < 0.01; *p < 0.05.

## Discussion

We sought to investigate neurocognitive mechanisms of consolidation during sleep using targeted memory reactivation of sets of either one, two, or six items. Results showed that TMR produced a clear benefit in the form of better recall for cued items than for non-cued items. Moreover, this memory benefit was not influenced by set size, ruling out the possibility that reactivation in our study can be conceptualized as a limited resource that is divided among individual memories (i.e., the LRH). Additionally, our data do not support the hypothesis that memory reactivation benefits only a small number of items within a set that are sampled either randomly or in a biased manner (e.g., weaker items more likely to be reactivated[8,25]). The random subset reactivation model predicts bigger benefits for smaller sets and a correlation between the number of times a cue was repeated during sleep and the average benefit for the associated set. Neither of these predictions were borne out. The biased subset reactivation model predicts higher within-set agreement between benefit scores for non-cued sets, yet our data showed no significant differences between ICC scores and outlier scores for cued and non-cued sets.

Taken together, our results support the PRH, suggesting that reactivation occurs in a parallel manner and that multiple memories can be reactivated independently at the same time. An alternative interpretation of our data could be that items were not reactivated simultaneously, but rather in quick succession. Although we cannot rule out this interpretation, we believe it to be less likely. Reactivation in quick succession would allow different items within a set to be sequentially and rapidly reactivated over several hundreds of milliseconds after cue presentation. Indeed, replay during sleep in rodents can be temporally compressed by a factor of 20[37,38] and recent evidence in humans also suggests temporal compression is at play, at least during wake[39,40]. However, studies exploring the timescale for reactivation in TMR suggest that at least some aspects of reactivation occur seconds after cue onset (e.g., EEG activity seconds after onset encompasses stimulus-related information and predicts memory benefits[35,41]; cuing benefits are eliminated if reactivation is disrupted <~1 s after onset[42,43]). These results are complemented by evidence from animal studies showing a cortico-hippocampal-cortical loop for spontaneous (i.e., not cue-driven) reactivation lasting several hundreds of milliseconds[44,45]. Taken together, these findings suggest that the full cycle of reactivation may last substantially longer than replay expressed by the hippocampal neuronal ensemble. If so, reactivation of multiple memories in sequence on those longer timescales should produce different temporal profiles (e.g., prolonged activity in the spindle or delta-theta ranges) for sets of different sizes, but our data do not support this notion. Our data therefore fit better with the notion of parallel reactivation rather than fast sequential reactivation. Further investigation should attempt to tease apart features of parallel versus fast sequential reactivation. Pursuing this line of inquiry could provide crucial information regarding the mechanisms and timescales of sleep-related memory reactivation.

Whereas our findings constitute a step forward in constructing a neurocognitive model for memory reactivation, they also raise new questions deserving of investigation. In the context of TMR, cue presentation first initiates primary sensory processing (i.e., processing that depends on stimulus properties and not the scope of the memories associated with it). What happens next is open for exploration. One possibility is that the stimulus representation

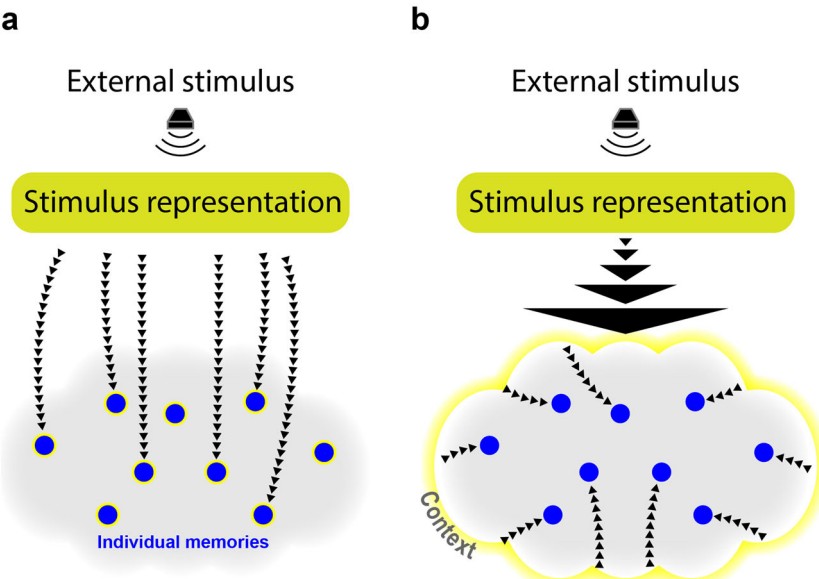

**Fig. 5 Models explaining the behavioral results, supporting the parallel reactivation hypothesis. a** In one model, individual memories are directly and simultaneously reactivated following stimulus presentation, regardless of the context in which they are embedded (e.g., locations of different cat images activated following a "meow" sound). Note that not all items in the context are reactivated (e.g., not all cat-related memories). **b** In contrast, in another model the stimulus reactivates a generalized context and its reactivation subsequently benefits individual memories embedded in the context.

is linked with multiple, independent, and highly-specific memory traces (e.g., the spatial position of a single item out of a set of multiple items), which are then reactivated in a parallel manner (Fig. 5a). The major characteristic of this model is parallel offline reactivation at the single-item level. To avoid contamination between different memories that are reactivated at the same time (e.g., swapping of memory-specific information that would effectively damage both memory traces), these reactivations must maintain some level of independence one from the other.

A potentially important property of our paradigm is that stimuli and sounds within sets had a close semantic relationship. It is possible that generalized representations of a set – and not only the representations of the individual items – may be of importance for sleep's effect. As an alternative to the item-reactivation model, it could therefore be suggested that the semantic theme of a set forms a context that is reactivated, benefitting all embedded items. This model is reminiscent of computational models of the effects of context on episodic memory, such as the Context Maintenance and Retrieval (CMR) model[46,47]. Briefly, this model predicts that memory search will be determined by associations between items and the context in which they are embedded. The context includes both a semantic clustering component and a temporal clustering component (i.e., items learned in temporal proximity will share the same context). Whereas the interplay between context and memory has been extensively studied, focusing both on its role in recall[46,48,49] and with regard to the role of the hippocampus in binding item memories to context[50,51], the role of context in sleep reactivation has not been systematically explored. If contexts are directly reactivated (whether spontaneously or using TMR), they may then reactivate individual embedded items (Fig. 5b). Note that mathematical models of memory posit competition between items linked to a context[52]; however, these models have focused on data from wake, not sleep, and it is possible that memory systems operate differently during sleep. For example, low levels of acetylcholine during slow-wave sleep may put the hippocampus into a strong retrieval mode that makes it possible to retrieve more traces at once[53,54] (but see Klinzing et al.[55] for results indicating the TMR benefits are independent of acetylcholine).

The idea that TMR cues activate contexts (as opposed to or in addition to directly activating items) may help to explain data on the time course of TMR. Bendor and Wilson[22] presented learning-related auditory cues during sleep and found a bias toward reactivation of cued memories, but intriguingly, this bias was sustained for multiple seconds after cue offset. At that point in time, any replay that was directly elicited upon initiation of the auditory cue should have ended. One of the characteristics of context, at least during wake, is its slow temporal drift[56], and therefore these data may be better fit by a model in which the TMR cue activates a context representation that can persist over time to impact individual items.

The notion of parallel processing of memories during sleep may in some ways resemble that of parallel cognitive processing during wake[57]. Specifically, it has been shown that multiple memory retrieval processes can overlap in time, but only if the retrieved memory items belong to the same category[58,59]. It is unclear whether the same mechanisms underlie parallel reactivation during wake and sleep and whether there are differences in how both affect memory consolidation. If parallel memory reactivation indeed occurs similarly during wake and sleep, this may imply that reactivation during sleep would also be limited to strongly linked items, such as those belonging to the same category.

To reveal neural correlates of reactivation of multiple items over sleep, we analyzed modulations in sigma and delta-theta frequency bands following cue presentation during NREM sleep. Both sigma power, which includes sleep spindle activity, and delta-theta power, including large slow waves and K-complexes, increased abruptly in the seconds following sound onset. Crucially, these boosts in activity were modulated by the size of the set associated with the presented sound, with the highest increase in both bands measured after sounds associated with six items. This pattern was also apparent when considering spindle probability over the same timeframe.

Spindles have long been associated with memory consolidation during sleep and have been hypothesized to embed previously learned information[36]. Recently, spindles have been shown to coincide with time windows in which decoding of stimulus

properties is possible[35], further solidifying their significance for memory consolidation. Previous studies have contrasted spindles following previously learned and novel stimuli[60], yet our results seem to be the first to manipulate memory load and consider its effect on spindle amplitude and probability, showing that the extent of previous learning affects these measures.

Whereas stimulus evoked power in the sigma range has been consistently linked with memory improvements, stimulus-evoked delta-theta power modulations have attracted less attention. Oscillations in the lower delta range (0.5–1 Hz) are thought to govern the cortico-hippocampal coupling necessary for consolidation[3]. Slow-wave activity during sleep has been shown to be higher after learning both globally[61] and locally for previously engaged brain regions[62], but these effects reflect a coarse, task-related increase that is not linked to any specific memory.

The other major waveform contributing to the spectral activity in the delta-theta range is the K-complex. Specifically, the sharp rise in amplitude associated with these generally slow (~1 Hz) waves extends its spectral signature to the theta range (4–8 Hz). K-complexes are usually explored in the context of sensory processing and have been hypothesized to protect against arousal[63], but have been linked to memory consolidation as well[64]. This gating is not purely sensory and depends at least to some degree on cognitive analysis, as evidenced by larger K-complexes to cognitively salient stimuli such as one's own name[65,66]. Indeed, there is growing evidence suggesting that induced activity in both the theta and delta ranges may reflect previous learning. Recently, Andrillon and colleagues[67] found that learning increases triggered delta power. Similarly, theta activity was also increased for sounds associated with previous learning relative to novel sounds[68]. Additionally, similarities between wake- and sleep-related theta activity after cue onset were found to predict subsequent memory, suggesting that theta activity may contribute to memory consolidation[69]. Our results seem to be the first showing that delta-theta power is correlated with the scope of information linked with a stimulus, further contributing to the hypothesis that delta-theta power is sensitive to information at higher levels than previously thought.

Our results showed a seemingly discrepant pattern: the behavioral finding of set-size-independent memory benefits on the one side and the physiological finding of set-size-dependent increases in delta-theta and sigma power on the other. However, the set-size-independent memory effect held only for the average benefit per item (averaged across one, two, or six items). When considering the cumulative benefit combined over all items in a set, there was a marginally larger cumulative benefit for larger sets. Delta-theta and sigma power may therefore represent the cumulative benefit for the set associated with the presented cue. Alternatively, the physiological measures may represent the process by which memories are retrieved and modified—in which case delta-theta and sigma power may reflect the extent of previous knowledge made available for consolidation. Either way, the physiological and behavioral findings can be reconciled based on these considerations and thus are not at odds with each other.

The PRH seemingly suggests a limitless capacity for simultaneous memory reactivation. However, we used relatively small sets in this study, and therefore cannot specify an upper boundary on reactivation capacity; certainly, sets larger than six items may show graded benefits or entail interference among memories. To date, only two studies have explored the capacity for reactivation during sleep, showing that high memory loads nullify sleep-related consolidation effects[13,14]. However, these studies are agnostic as to whether consolidation happens sequentially or simultaneously.

Finally, some limitations must be acknowledged. First, our sample consisted mostly of university students. More research is needed to establish the generalizability of these results and test the possibility that population-wide variability in cognitive functions (e.g., working memory[70]) may predict an individual's capacity for simultaneous reactivation. Another limitation of the current study is that we cannot address whether our findings were influenced by our choice of stimulus categories (i.e., similar and highly associated items with the same congruent sound, that may have been chunked together to constitute a single, unitized memory), or by other design features that could limit generalization to other situations. For example, Antony et al.[19], using a different spatial-TMR procedure, found that benefits for items within a cued-pair were anti-correlated, suggesting there may be conditions in which only a subset of items are reactivated (but see contrasting results in Vargas et al.[20]). Differences across designs do not allow direct comparisons with our results, so explanations for these discrepancies remain outstanding.

Another related point concerns variability in relatedness between different sets that were used in our study (e.g., it could be that the "Cat" set was more interrelated than the "Airplane" set). Importantly, these differences should not have biased our results in any way, although they may have introduced noise and therefore decreased power. The selection of which sets were cued and which were not was accomplished with an algorithm to effectively balance between-set memory differences. Our results open the door to further investigation of the importance of relatedness to sleep-related memory consolidation. Future studies should attempt to manipulate item relatedness to reveal the boundary conditions of simultaneous reactivation during sleep.

In an effort to avoid confounding differences between sets of different size, including factors such as relatedness, we randomly assigned, per participant, each multi-item set to either the six- or two-image condition (see Methods section). Ideally, we would have randomly assigned sets across all three conditions (i.e., six-, two-, and one-item sets). However, we were not able to find enough image categories that include six distinct images without substantially sacrificing stimulus quality (e.g., choosing groups that were too similar to other groups or including confusable images). We cannot therefore rule out that this design weakness had some effect on our results. However, the patterns observed in our results across set sizes (e.g., the lack of an effect of set size on cuing benefits) do not seem to be limited to multi- vs. single-item sets, but rather seem to also be manifested when comparing the well-balanced six- and two-item sets. This lends support to our claim that stimulus factors did not substantially threaten the validity of our results.

The growing appreciation of the importance of sleep for memory consolidation has raised unexplored questions regarding the relevant neurocognitive mechanisms. Our results suggest that reactivation of multiple memories can occur simultaneously and independently during sleep, benefiting several memories in parallel. Reactivation capacity during sleep may therefore be larger than would be assumed with serial reactivation, as in typical working-memory operations. Although our data are consistent with models in which single memory items are reactivated directly, an alternative, previously unexplored model in which generalized contexts are reactivated seems equally likely and worthy of further exploration. Either way, the notion that reactivation benefits memories in a parallel and promiscuous manner suggests that the capacity for sleep-related consolidation is far beyond that which was previously assumed.

## Methods

**Participants**. Participants had no known history of neurological or sleep disorders and claimed to be able to nap in the afternoon. In preparation for the study, participants were asked to go to bed later than usual the night before the study, wake up earlier in the morning, and avoid any caffeine on the day of the study. Our

sample included 40 participants, but nine of these were excluded from the study because they were not exposed to each of the to-be-cued stimuli during sleep. The final sample included 31 participants (9 males, 21 females, and one non-binary person) between the age of 18 and 30 years (mean ± SD = 20.81 ± 2.96). The Northwestern University Institutional Review Board approved the procedure.

**Materials**. Visual stimuli were presented on a screen (1920 × 1080 pixels, P2418HT, Dell Inc., TX). Sounds were delivered using speakers (AX-210, Dell Inc., TX). Stimulus presentation was controlled by Neurobs Presentation (v17.2).

Visual stimuli consisted of images of objects, parts of objects, or people, each shown at 125 × 125 pixels (34.6 × 34.6 mm). A total of 45 sets of pictures that shared the same theme were used (e.g., images of different cats; different parts of an airplane; Supplementary Table 1). Each set consisted of main images for the spatial-memory task and lure images only needed for the item recognition task, as described below. Twenty-four of the sets included 6 main images and 12 lure images ("multi-item sets"). Twenty-one of the sets included one main image and two lure images ("single-item sets"). Each set was matched with a single, distinguishable, congruent sound with a maximal duration of 600 ms (e.g., a meow; a take-off sound).

The 24 multi-item sets were randomly assigned, per participant, to 18 two-item sets and 6 six-item sets. The rationale behind this random assignment was to avoid any systematic differences between six- and two-item sets that were independent of set size. For each of the two-item sets, two main images and four lure images were randomly chosen to be part of the set. Three of the single-item sets were practice sets that were used for initial training and also acted as fillers in pre- and post-sleep tests. A single, 500-ms sound not associated with any item was only presented during sleep ("novel sound").

Together, the 18 one-item sets, 18 two-item sets, and 6 six-item sets consisted of 90 different images. For the spatial task, each of these images was paired with a single location on a circular grid with a radius of 540 pixels (149.4 mm). The position for the center of each image was randomly selected to obey the following rules: (1) it was at least 50 pixels from the center of the grid; (2) it was at least 50 pixels from the external border of the grid; (3) it was at least 41 pixels from the location of any of the other 89 items; and (4) it was at least 400 pixels from any other item in its own set. The rational for this last rule was to allow us to disentangle swap errors and accuracy errors (see below). However, because this rule is applied differently to sets of different size, changing the distribution of potential locations in a set-dependent manner, we randomly generated sets of six locations for each set, regardless of its real size. We then assigned all six locations to items in six-item sets and randomly chose two locations or one location for the two- and one-item sets, respectively.

The 90 items were split into six learning blocks of 15 items, so that items of the same set were never learned in the same block. Items belonging to sets of two items were always learned in sequential blocks (i.e., blocks 1–2, 2–3, 3–4, 4–5, or 5–6).

**Procedure**. After consenting to the study, the participant was fitted with an electroencephalography (EEG) cap. Next, a test was administered to measure pre-sleep response time (RT) in order to later evaluate sleep inertia (as described below). This RT task consisted of a red square that shifted between left and right positions at 10 Hz and finally stopped at one of the two locations. The participant was required to click the correct mouse button (i.e., left/right) before the square began flickering again. The task ended when the participant responded correctly for 8 out of the last 10 trials. Initially, the square paused for 450 ms, but if the participant failed to reach the criteria within 30 trials, this duration was extended by 50 ms and the task restarted, iterating until the criterion was reached.

The RT task was followed by instructions and practice trials for the spatial-memory task. This task consisted of six blocks, each including 15 items. Each block started with exposure trials, in which the grid appeared with a sound. One second later, a corresponding image appeared in its location. The image disappeared 3 s later, coinciding with the offset of another presentation of the sound. Two seconds later, the grid disappeared, followed by a 1-s white-screen inter-trial interval (ITI), progressing through all 15 items.

Next, the positioning trials began. Each of these trials started with the simultaneous presentation of the circular grid, the item, and the related sound. The item was positioned in a random location at least 100 pixels from the correct item location. The participant used the mouse to drag and drop the item in its correct position. Trials were self-paced and every time an item was picked up or dropped the associated sound was presented. The participant signaled their choice by clicking the right button, which triggered a 3-s feedback screen with the image at its correct location. Some movement from the initial position was always required, and placements were considered correct if <100 pixels (27.7 mm) from the correct location. For incorrect placements, a red arrow linked the participant's choice with the correct location. After feedback, a 1-s white-screen ITI commenced, followed by the next trial.

Positioning trials ended when the participant responded correctly for each item twice in a row. When this criterion was reached for an item, it was no longer included. Items were presented pseudo-randomly, with the only limitation being that items were not displayed twice in a row (unless only a single item remained).

After all six blocks were completed, the pre-sleep test began. It consisted of positioning trials for all 90 items (plus the three practice items, which were

presented first). These positioning trials were identical to the positioning trials during training, except that no sounds were presented and no feedback was provided. The order of items in this stage was pseudo-random, with any two items belonging to the same set separated by at least two items from a different set.

Based on the results from this stage, half of the sets were chosen to be cued during sleep. Across all six-item sets, those chosen to be cued were balanced with the remaining sets by first minimizing differences between the numbers of incorrectly placed items. If those were equal between sets, the number of swap errors and the absolute error in pixels (see definitions below) were also balanced. The same procedure was also used for two-item sets and one-item sets.

Next, the futon chair on which the participant was seated was converted on a bed. During the ensuing 90-min nap opportunity, the participants' EEG, electrooculography (EOG), and electromyography (EMG) were continuously monitored. White noise was presented over a set of speakers positioned ~200 cm from the participant's head. When the participant entered slow-wave sleep (stage N3) cuing ensued. In total, 22 sounds were presented to each participant: three six-item sounds (out of six), nine two-item sounds (out of 18), nine one-item sounds (out of 18), and one novel sound. All 22 sounds were unobtrusively presented in a random order, and then the sequence was re-randomized and presented again. This method was used to balance the number of times sounds associated with different set sizes were presented. Sounds were presented until the participant showed signs of arousal. The inter-cue interval was randomly chosen per presentation to be either 4.5, 5, or 5.5 s. If cuing was not completed after 45 min and the participant was not in stage N3, cues were also presented during stage N2. Variation in the duration of NREM between participants resulted in differences between the number of repetitions per sound (range: 1–24; see Fig. 3c).

Out of 31 participants, five reported upon awakening that they heard sounds during sleep. At the very end of the task, they were presented with all the sounds and asked to specify for each whether they heard it during sleep or not. Importantly, these five participants were all at chance level (1.4 ± 0.51 hits; 2.2 ± 0.66 false alarms out of 43 sounds).

After waking up, the bed was converted back into a chair. Testing commenced at least 5 min after sleep offset. The participant was first required to reach criterion in the RT task to assure sufficient alertness. The time-window for response was based on their own responses in the pre-sleep RT task. Only one participant had trouble reaching that criteria, and was given an opportunity to freshen up before trying again, subsequently succeeding to reach criterion.

The participant then started a post-nap spatial-memory test, which was identical to the pre-sleep test. This testing was followed by an item-recognition task, in which the participant heard the sound for each set along with a set of images and had to indicate for each image whether it was previously presented as part of the spatial-learning task or not. For each old image, two lures of the same semantic category were presented. Each image was presented twice. Data from this task were not used for this manuscript.

For debriefing, participants were asked whether they heard any noises during the nap and what they thought about the purpose of the study. They were then paid and dismissed.

**Electroencephalography and polysomnography**. EEG was recorded using Ag/AgCl active electrodes (Biosemi ActiveTwo, Amsterdam). In addition to the 64 scalp electrodes, contacts were placed on the mastoids, next to the eyes, and on the chin. All recordings were made at 512 Hz. Scoring was approximated online while the participant was sleeping to determine when to commence and cease sound presentation, and later completed offline using the EEGLAB[71] and sleepSMG (http://sleepsmg.sourceforge.net) packages for Matlab 2016b (MathWorks Inc., Natick, MA). EEG channels were re-referenced to averaged mastoids and filtered using a two-way least-squares FIR bandpass filter between 0.4 Hz and 60 Hz (*pop_eegfilt* function in EEGLAB). Noisy channels were replaced with interpolated data from neighboring electrodes using the spherical interpolation method in EEGLAB. Online and offline sleep scoring were both based on the guidelines published by the American Academy of Sleep Medicine[72]. Offline scoring was done by two independent raters, both of whom were not privy to when sounds were presented. Any discrepancies were subsequently reconciled by one of the two raters. For all analyses, only cues that were presented during NREM sleep (N2 and N3) were considered. Of these, an average of 84.8% (across participants) were presented during N3.

**Statistics and reproducibility for behavioral results**. Analyses consisted of ANOVAs, repeated-measures ANOVAs, paired and non-paired t tests, Tukey post-hoc highly significant differences (HSD), correlation analyses, and interclass correlation (ICC) analyses, as described in the main text. Analyses were completed using Matlab 2016b (MathWorks Inc, Natick, MA) and SPSS 25 (IBM Inc., Armonk, NY). Effect sizes are presented using partial $\eta^2$.

For the spatial-memory task, we define two types of placement errors: swap errors and accuracy errors. Swap errors are errors in which a participant mistakenly positioned an image near the location of another image belonging to the same set. If an item was positioned closer to another same-set item's position, it was considered swapped. Accuracy errors, measured in pixels, reflect the distance between item placement and its correct position. Our design allowed us to disentangle swap errors and accuracy errors (i.e., correct item locations within a set

were far apart, as described above). For each item placement in the pre- and post-nap tests, we first considered whether there was a swap error. Only non-swapped items were considered for the accuracy error analyses. The rationale behind detecting and removing swap errors from accuracy estimates is that they inflate errors in a manner that does not purely reflect spatial-memory.

A caveat of our method of detecting swap errors is that pure guesses may also be counted as swap errors if they happen to be closer to another image location than to the correct one. We were aware of this limitation, but preferred to overestimate rather than under-estimate swap errors to avoid their contamination of accuracy errors. However, our definition of swap errors could create an unfair advantage in error rate for bigger sets. For example, a large error due to guessing (e.g., 800 pixels) would more likely have been closer to some other item in a six-item set and be counted as a swap error relative to a two-item set. The accuracy error for this item would therefore not have been considered for the larger set. The same error for an item in a single-item set would not have another item to swap with and the accuracy error would have been considered, inflating errors for smaller sets. To compensate for this bias, swap errors were calculated by taking into consideration six locations, even for sets of one or two items. For example, a swap error in a one-item set would have taken place if the participant mistakenly positioned the single item closer to any of five other locations on the gird, despite the fact that these locations were never linked to any item. These alternative locations (five alternatives for the one-item set and four for the two-item sets) were at least 400 pixels apart from each other and from the correct locations of the items in the set. All swapped items for all set sizes were removed from the accuracy error analyses. Note that larger sets would still have more identified swap errors, because this value encompassed both large guessing-related errors (which are presumably identical for items of different set sizes) and errors due to item swapping (which should be higher for larger sets). The average number of swap errors, which is equivalent to the number of items removed, is presented in the Results section for the two- and six-item sets. The average number of swap errors for one-item sets was $0.1 \pm 0.01$ before sleep and $0.09 \pm 0.02$ after sleep.

The benefit of sleep was calculated for each item as the accuracy error in the post-nap spatial-memory test subtracted from the error in the pre-sleep test (i.e., positive values represent memory improvement). For the basic analysis of the effects of cuing and set size (Fig. 2), item benefits were averaged for each set size and cuing status within participant. In addition to this per-item analysis, we also conducted analyses for cumulative benefits within sets (e.g., the added cuing benefit within a set of six items). To confirm that our per-item results were not due to differences in pre-sleep spatial error rates between sets of different sizes, we complemented our main analysis with two additional ones (Supplementary Fig. 2). For the first analysis, we regressed out pre-sleep error rates by calculating the linear relationship between pre-sleep errors and forgetting (post-sleep – pre-sleep spatial accuracy). Then we subtracted each spatial forgetting score from the spatial forgetting expected from this linear relationship (i.e., the residual) and added back the mean raw spatial forgetting value to produce an adjusted accuracy error score. The results obtained using this corrected score are presented in Supplementary Fig. 2a.

For the second analysis, 50% of trials were randomly chosen and considered for analysis for each participant and each set size. We next considered the pre-sleep accuracy error rates (averaged across individual items) for the datasets generated using this method, and eliminated all datasets in which the set size main effect had $p < 0.5$. Using this method, we collected 500 subsampled datasets in which the pre-sleep data was negligibly affected by set size. Finally, we ran a repeated-measures ANOVA to calculate the cuing effect and the cuing by set size interaction for these 500 datasets. The $p$-values and effect sizes obtained by these ANOVAs are presented in Supplementary Fig. 2b.

For the ICC analysis (Fig. 3a), significance of the difference between the benefit for cued and non-cued sets was assessed by applying a permutation test. The ICC was first computed for cued and non-cued sets separately and the difference between these measures was calculated. Then, the data was shuffled $10^6$ times, so that the assignment of each individual item benefit to a specific set was random in each permutation. The shuffling was done separately for the cued and non-cued sets and the ICC was calculated for both. Then, $p$-values were assessed by comparing differences between real ICC values with the distribution of random differences.

**Statistics and reproducibility for spectral analyses and analysis of sleep spindles**. Analyses were completed using Matlab 2016b (MathWorks Inc, Natick, MA), including the EEGLAB[71] package. For the spectral analysis, each participant's continuous sleep data was segmented into epochs starting 500 ms and ending 4000 ms after cue onset. Spectrograms were calculated using short-time Fourier transform of the mean-subtracted segmented data (*spectrogram* function in Matlab). Data were obtained for frequencies ranging between 0.25 Hz and 25 Hz in 0.25 Hz steps using a 500-ms sliding window with 87.5% overlap. Analyses were conducted on data obtained from electrode Cz, but spectrograms for all electrodes are presented in Supplementary Fig. 3a.

To identify reliable clusters of activity, we averaged the time-frequency responses to all sounds for each participant and then calculated the percentage of change in power relative to a 300-ms interval starting 500 ms before cue onset (Fig. 4a, top). We then conducted a $t$-test across subjects to detect points in the

time-frequency space in which power was systematically different from zero ($p < 0.001$, Bonferroni corrected; Fig. 4b, bottom). We identified two large clusters of significant points in the time-frequency response: one in the delta and theta ranges and another in the sigma range (scalp distributions for both presented in Supplementary Fig. 3b).

Using the two identified clusters, the delta-theta cluster and the sigma cluster, we calculated a baseline-corrected average modulation across time and frequency for each trial and grouped them according to set size (Fig. 4b). To estimate the effect of the number of items associated with each sound on spectral power, we used linear mixed models with the number of items (i.e., zero, one, two and six) as a fixed effect. Two random effects were accounted for: participant (i.e., to account for random differences between participants) and sound identifications (i.e., to account for random differences between sound-specific EEG responses within participant). The models we used is therefore the following:

$$\text{Modulation} \sim 1 + \text{SetSize} + (1 + \text{SetSize}|\text{SubjectID}) + (1|\text{SoundID})$$

where Modulation is the change in sigma or delta-theta power. This nested model produced an estimate of the linear coefficient for the delta-theta and sigma clusters (Fig. 4c). The obtained modulation coefficient reflects the change in induced power (in percent, relative to baseline) with every additional item.

Sleep spindles were automatically detected using custom Matlab scripts. Periods of time which included artifacts (e.g., movement during sleep) were manually detected and omitted from analysis. All segments of data from electrode Cz during artifact-free periods of NREM sleep were used for analysis. The EEG signal was filtered between 11 and 16 Hz, and then the root mean square (RMS) was calculated at every time point with a moving window of 200 ms. Spindles were detected when the RMS crossed and remained above a threshold of 1.5 standard deviations of the signal for 500–3000 ms. The detected spindles were segmented around cue onsets. The same 300-ms interval was used for baseline correction for spindle probabilities (Fig. 4d, e). The peak amplitude for each spindle was defined as the maximum between the amplitude at the highest peak and the absolute value of the amplitude at the lowest trough within the spindle timeframe. A linear mixed model was used to estimate the effect of the number of items on the spindle probability and peak spindle amplitude. Details of this analysis (e.g., fixed and random variables) were as described for the power modulation.

Finally, another mixed linear model was used to reveal the effects of the physiological modulations on both per-item and per-set cuing benefits. For this, the following model was used:

$$\text{Memory Benefit} \sim 1 + \text{Modulation} \times \text{SetSize} + (\text{Modulation}|\text{SubjectID})$$

## Data availability
The datasets generated during the current study are available from the corresponding author on reasonable request. Source data for all figures is available in Supplementary Data files.

## Code availability
The scripts used for data collection (Neurobs Presentation, v17.2) and analysis (Matlab 2016b) are available from the corresponding author on reasonable request.

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

## Acknowledgements
This work was supported by NSF grants BCS-1461088, BCS-1829414, and BCS-1533511. E.S. was funded by the Human Frontier Science Program and the Zuckerman STEM Leadership Program.

## Author contributions
E.S., J.W.A., K.A.N., and K.A.P. designed the study. E.S., B.J.W., and A.L. assembled the stimulus sets, and collected the data. E.S. analyzed the data and wrote the first draft of the manuscript. All authors (E.S., J.W.A., A.L., B.J.W., K.A.N., and K.A.P.) edited the manuscript.

## Competing interests
The authors declare no competing interests.
