## [Peer Review File · Communications Biology]

Reviewers' comments:

Reviewer #1 (Remarks to the Author):

This was a strong paper. It was clearly written, provided useful illustrations, and had a strong theoretical framework. The hypotheses were well-described, the results were generally compelling, and the limitations were acknowledged. The application of context-based models for understanding parallel reactivation seems an important advance. I don't have any major concerns with the manuscript. I do have a few queries for the authors that I believe will increase the strength and consistency of the exposition.

Literature Review

1. One area the authors can strengthen is incorporating what is known about parallel reactivation while in the wake state. There's a small literature on this topic, and a useful starting point would be Rohrer, Pashler, and Etcheagaray's work (1998, *Memory & Cognition*). The authors will likely find similarity in Rohrer's findings on simultaneous retrieval for semantic categories, given that the current study reported simultaneous reactivation for materials that were semantically grouped. In addition, I would be curious to know whether the authors view the parallel reactivation during sleep as more akin to semantic priming reactivation or episodic memory retrieval, which might inform both capacity questions (see comment #2) and the extent to which memories are simultaneously reactivated versus reactivated in quick succession (see comment #5).

Exposition

2. While I am generally convinced by the authors' data that more than one item memory can be reactivated simultaneously, I have questions about the generalizability of this effect—particularly in considering whether reactivation is limited vs limitless in capacity. The authors acknowledged that graded effects could occur with larger sets in the "limitations" paragraph, but this issue strikes me as more fundamentally important than to call it a limitation and move on. The notion that parallel reactivation can occur, but that it has some constraints (e.g., isolated to one category, bound to a specific context, etc.), is worthy of its own discussion paragraph. Doing so will advance the parallel reactivation account from what's known to what seems logical but still needs testing. Graded effects, at some point, seem like a logical necessity.

3. As mentioned above, I thought the context-based explanations of TMR were great. This framework also seems to better explain why TMR not only benefits item memory but also can benefit integration of items (due to simultaneous reactivation of different items).

4. I assume that the participants were undergraduate students at Northwestern. How might the high intellectual capacity of such individuals impact the finding of no decline with increasing set size, and subsequent theoretical conclusions? In other words, is it a characteristic of high intellectual capacity individuals that they can reactivate memories in parallel in sleep, or is this a characteristic of all 18-30 year olds? There's at least some evidence – from a state university with greater intellectual variability in its student participants – that higher working memory capacity predicts greater memory retention across sleep intervals (Fenn & Hambrick, 2012, *JEPG*).

5. On p. 15, the theoretical framing took a subtle turn. Up to this point, the framing had focused on

simultaneous reactivation of memories—a very provocative idea—but on p. 15 switched to the more established notion of reactivations occurring in quick succession. The quick-succession account would be a non-simultaneous reactivation account. Further thought, and more cautious wording, is warranted here (or throughout, if the authors believe quick succession better captures their findings than a purely simultaneous reactivation).

Methods

6. The loss of 22.5% of participants (9 of 40 participants) was a limitation. The authors indicated that these participants were not exposed to each of the cued stimuli. Is that because they woke up too early? What percentage of items were they exposed to, and if they were exposed to most of the items (>75%), couldn't the analyses focus on proportion of total items exposed to retain more participants?

7. The main text and methods section could be clearer in defining swap errors. Some revision would be welcome, perhaps including a supplemental figure to illustrate a swap error.

Signed,
Michael Scullin

Reviewer #3 (Remarks to the Author):

COMMSBIO-20-1738-T

Multiple memories can be simultaneously reactivated during sleep as effectively as a single memory

Schechtman, E., Antony, J. W., Lampe, A., Wilson, B. J., Norman, K. A. & Paller, K. A.

This study examined whether the effect of cued reactivation during sleep on memory depends on the amount of information being reactivated via cues during sleep. It aimed at elucidating whether cued reactivation of multiple memories entails costs. The study further examined the contribution of sleep-related brain oscillations in processing reactivated memories. One group of 31 participants encoded 90 picture-location-sound associations, which were grouped in 6-, two- or one-pictures-sets of different themes (e.g. 6 different cats, two different parts of airplanes). Each picture was associated with a specific theme sound (e.g. all cats – meow). After learning the positions of the pictures to criterion, participants slept and were presented with half of the sounds and one new sound during non-REM sleep (mostly SWS, very few S2), in order to reactivate single or multiple memories. Afterwards, correct placement of the pictures was tested. Results showed that set size did not modulate the success of reactivation for placement memory, but reactivated sets were overall recalled better than not reactivated sets. Cueing affected the delta-theta and the sigma frequency bands. Larger sets had a stronger modulatory impact on these bands with higher spindle probability associated with larger set size. But delta-theta and sigma frequency bands and set size were not associated with memory performance.

Overall, these findings are novel and highly interesting to the field of cued reactivation of memories during sleep in showing that the success of cued reactivation is not attenuated by the reactivation of multiple items compared to a single item and that set size during learning is associated with

oscillations in the sigma and delta frequencies, while these frequencies are not related to memory performance.

The methods are sound, statistics are elaborated and convincing, the manuscript is well-written and results are discussed appropriately. I have only a few comments that should be addressed.

Major points:

1. My first concern relates to the prediction of the cueing effect by physiological measures. In the discussion section, the authors mention that the cumulative benefit over items for the behavioral analysis is modulated by set size (see Discussion section, p. 19, line 422). Please add this analysis to the results section. Beyond that, do delta-theta power, sigma power and spindle probability predict this cumulative measure of cueing benefit (there is no such effect when predicting the averaged cueing benefit, see Results section, p. 14, lines 305 - 309). Please, discuss the (lacking) effect of physiological measures on (cumulative) cueing benefits.

2. My second point relates to the semantic relatedness of the stimuli within a set. It seems that different cats are related more closely than different parts of an airplane (example see Methods section, Materials, p. 22, line 472). Was semantic relatedness equally distributed among set sizes and among cued and not cued sets (i.e. among two- and six-item sets)?

3. Related to my previous point, do the authors expect any effects of semantic relatedness on their results? E.g., cued reactivation might be more effective for highly related sets if cued reactivation occurs in a generalized context manner, while it might be independent of semantic relatedness if cued reactivation occurs for individual memories. Please consider discussing this possibility and if suitable, include results.

Minor points:

1. As far as I understand, the ANOVA testing the effect of set size and cueing on change in swap errors included the main effect of set size. For the sake of completeness, please report results of this main effect (see Results section, p. 7, lines 141 – 146). The same accounts for the ANOVA testing the effect of the same factors on change in spatial errors controlled for pre-sleep errors (see Results section, p. 8, line 170-172) and for caption of Figure 2 (see Results section, p. 40, “The lower panel shows...”). Also, consider reporting the effect of these factors on pre-sleep swap error, since pre-sleep errors were corrected for swap errors and differ depending on set size.

2. Standard errors should be added to the averaged spatial-memory errors for cued and non-cued pictures (see Results section, p. 8, line 164).

3. The reference to Figure 4c) is missing in the description of the results of the linear mixed model (see Results section, p. 13, lines 279 - 284).

4. It is not clear to me, what exactly is meant by the random effect of sound identification in the description of the linear mixed model estimating the effect of set size on spectral power (see Methods section, p. 30, line 654). Please clarify.

5. Please, specify the linear mixed model to predict cueing benefit (see Results section, p. 14, lines 304 – 310).

6. Figure 2: please, add * to indicate significant results.

Dear reviewers,

We are pleased to present the revised version of our previous manuscript, titled “**Multiple memories can be simultaneously reactivated during sleep as effectively as a single memory**”. This version includes several changes motivated by your comments.

These improvements include new analyses and results (e.g., considering the cumulative benefits of reactivation within sets), an extended Discussion section (e.g., linking our results to parallel processing during wake; addressing issues related to within-set item relatedness), a revised Methods section (including clarifications regarding swap errors and mixed linear models) and a more in-depth consideration of the limitations of our study.

We are extremely grateful to the reviewers for their efforts and for recommending these various improvements. We hope they will find the revised version of our manuscript suitable for publication in *Communications Biology*.

Best,

Eitan Schechtman

In the name of all authors

Reviewer #1 (Remarks to the Author):

Literature Review

1. One area the authors can strengthen is incorporating what is known about parallel reactivation while in the wake state. There's a small literature on this topic, and a useful starting point would be Rohrer, Pashler, and Etcheagaray's work (1998, Memory & Cognition). The authors will likely find similarity in Rohrer's findings on simultaneous retrieval for semantic categories, given that the current study reported simultaneous reactivation for materials that were semantically grouped. In addition, I would be curious to know whether the authors view the parallel reactivation during sleep as more akin to semantic priming reactivation or episodic memory retrieval, which might inform both capacity questions (see comment #2) and the extent to which memories are simultaneously reactivated versus reactivated in quick succession (see comment #5).

We thank the reviewer for pointing out this fruitful direction. We have now added a paragraph in our discussion linking our findings to the broader literature on parallel processing (changes underlined).

The notion of parallel processing of memories during sleep may in some ways resemble that of parallel cognitive processing during wake⁵⁸. Specifically, it has been shown that multiple memory retrieval processes can overlap in time, but only if the retrieved memory items belong to the same category^{59, 60}. It is unclear whether the same mechanisms underlie parallel reactivation during wake and sleep and whether there are differences in how both affect memory consolidation. If parallel memory reactivation indeed occurs similarly during wake and sleep, this may imply that reactivation during sleep would also be limited to strongly linked items, such as those belonging to the same category.

Exposition

2. While I am generally convinced by the authors' data that more than one item memory can be reactivated simultaneously, I have questions about the generalizability of this effect—particularly in considering whether reactivation is limited vs limitless in capacity. The authors acknowledged that graded effects could occur with larger sets in the “limitations” paragraph, but this issue strikes me as more fundamentally important than to call it a limitation and move on. The notion that parallel reactivation can occur, but that it has some constraints (e.g., isolated to one category, bound to a specific context, etc.), is worthy of its own discussion paragraph. Doing so will advance the parallel reactivation account from what's known to what seems logical but still needs testing. Graded effects, at some point, seem like a logical necessity.

We agree that this point required further discussion. We have now added the following paragraph to our discussion (changes underlined):

The PRH seemingly suggests a limitless capacity for simultaneous memory reactivation. However, we used relatively small sets in this study, and therefore cannot specify an upper boundary on reactivation capacity; certainly, sets larger than six items may show graded benefits or entail interference among memories. To date, only two studies have explored the

capacity for reactivation during sleep, showing that high memory loads nullify sleep-related consolidation effects^{13, 14}. However, these studies are agnostic as to whether consolidation happens sequentially or simultaneously.

3. As mentioned above, I thought the context-based explanations of TMR were great. This framework also seems to better explain why TMR not only benefits item memory but also can benefit integration of items (due to simultaneous reactivation of different items).

4. I assume that the participants were undergraduate students at Northwestern. How might the high intellectual capacity of such individuals impact the finding of no decline with increasing set size, and subsequent theoretical conclusions? In other words, is it a characteristic of high intellectual capacity individuals that they can reactivate memories in parallel in sleep, or is this a characteristic of all 18-30 year olds? There's at least some evidence – from a state university with greater intellectual variability in its student participants – that higher working memory capacity predicts greater memory retention across sleep intervals (Fenn & Hambrick, 2012, JEPG).

We agree that our sample, consisting of mostly undergraduate Northwestern students, does not reflect the full range of intellectual capacity. We also acknowledge that there might be variability in the population with regard to sleep consolidation efficiency and that it may be related to other measures of memory, as reflected in Fenn & Hambrick (2012). We addressed the reviewer's comment in two ways. First, we now address this limitation in the manuscript (changes underlined):

Finally, some limitations must be acknowledged. First, our sample consisted mostly of university students. More research is needed to establish the generalizability of these results and test the possibility that population-wide variability in cognitive functions (e.g., working memory⁷¹) may predict an individual's capacity for simultaneous reactivation.

Second, we ran an additional analysis to consider whether inter-individual differences in memory within our sample predict higher benefits for items in smaller groups vs. items in larger ones. Our measure of spatial learning in no way measures intelligence, but should reflect, at least to some degree, cognitive-related individual differences. If participants with higher memory capacity are more capable of simultaneous reactivation during sleep, we would predict that error rates would be positively correlated with {benefit_for_set_of_1 - benefit_for_set_of_6} (e.g., a participant who minimized error rates would have similar benefits regardless of set sizes, and a participant who had large errors would have higher benefits for smaller sets compared to bigger ones). Our data show no significant correlation between these two measures ($r = -0.12$, $p = 0.51$; See figure below).

5. On p. 15, the theoretical framing took a subtle turn. Up to this point, the framing had focused on simultaneous reactivation of memories—a very provocative idea—but on p. 15 switched to the more established notion of reactivations occurring in quick succession. The quick-succession account would be a non-simultaneous reactivation account. Further thought, and more cautious wording, is warranted here (or throughout, if the authors believe quick succession better captures their findings than a purely simultaneous reactivation).

We thank the reviewer for this helpful comment. The point we were trying to make was that this explanation is less likely. We have reworded the paragraph to better reflect our position (changes underlined):

An alternative interpretation of our data could be that items were not reactivated simultaneously, but rather in quick succession. Although we cannot rule out this interpretation altogether, we believe it to be less likely. Reactivation in quick succession would allow different items within a set to be sequentially and rapidly reactivated over several hundreds of milliseconds after cue presentation. Indeed, replay during sleep in rodents can be temporally compressed by a factor of 20^{37, 38} and recent evidence in humans also suggests temporal compression is at play, at least during wake^{39, 40}. However, studies exploring the timescale for reactivation in TMR suggest that at least some aspects of reactivation occur seconds after cue onset (e.g., EEG activity seconds after onset

encompasses stimulus-related information and predicts memory benefits^{35, 41}; cuing benefits are eliminated if reactivation is disrupted (<~1 s after onset^{42, 43}). These results are complemented by evidence from animal studies showing a cortico-hippocampal-cortical loop for spontaneous (i.e., not cue-driven) reactivation lasting several hundreds of milliseconds^{44, 45}. Taken together, these findings suggest that the full cycle of reactivation may last substantially longer than replay expressed by the hippocampal neuronal ensemble. If so, reactivation of multiple memories in sequence on those longer timescales should produce different temporal profiles (e.g., prolonged activity in the spindle or delta-theta ranges) for sets of different sizes, but our data do not support this notion. Our data therefore fit better with the notion of parallel reactivation rather than fast sequential reactivation. Further investigation should attempt to tease apart features of parallel versus fast sequential reactivation. Pursuing this line of inquiry could provide crucial information regarding the mechanisms and time-scales of sleep-related memory reactivation.

Methods

6. The loss of 22.5% of participants (9 of 40 participants) was a limitation. The authors indicated that these participants were not exposed to each of the cued stimuli. Is that because they woke up too early? What percentage of items were they exposed to, and if they were exposed to most of the items (>75%), couldn't the analyses focus on proportion of total items exposed to retain more participants?

We thank the reviewer for expressing this concern. Presenting auditory stimuli during sleep sometimes causes arousal. Signs of arousal are monitored throughout the study, and auditory cuing is stopped immediately. For most participants, these arousals are relatively rare, allowing the experimenter to present all required sounds throughout the 90-minute session. However, some participants are highly arousable and remain easily arousable throughout the nap. In this study, for these participants, we temporarily lowered the volume of the sounds to allow participants to adapt to the sound presentation before gradually increasing the volume to above our pre-set threshold. In our analysis, we only considered sounds that were above this threshold and omitted all sounds that were below it. As noted in the manuscript, we only included participants who were exposed to each sound at supra-threshold volume at least once during stage N2 or SWS. Altogether, eight participants did not meet this criterion. Out of the 22 presented sounds, these participants were exposed to the following number of sounds: 0, 3, 3, 4, 7, 9, 13, 16 (another participant decided to withdraw from the task before the nap portion). Since there are relatively few cues that were properly presented for these participants (<75%), we decided that it would be misleading to include these participants in any analysis.

In general, a dropout rate of 22.5% is not unusual for auditory TMR studies (see for example Cairney et al., Sleep, 2017 (45% dropout); Simon et al., Neurobiology of Learning and Memory, 2018 (25% dropout); Wang et al., Journal of Neuroscience, 2019 (55% dropout)). The combination of sleeping with EEG electrodes in a new environment, together with sound presentation, increases the likelihood of participants waking up. However, the reviewer's statement is correct - if a large portion of the sounds were presented, it would have made sense to analyze these data. Unfortunately, this was not the case for this subset of participants. Our data collection plan was to continue running participants up until we reached at least 30 usable participants, and the omission rates were monitored during the data collection stage to ensure that we will reach this goal. Therefore, we do not feel that the

power of the study was compromised by omitting the participants and do not think analyzing their data would contribute to the manuscript.

7. The main text and methods section could be clearer in defining swap errors. Some revision would be welcome, perhaps including a supplemental figure to illustrate a swap error.

We thank the reviewer for pointing this out. We have revised and extended the sections discussing swap errors in both the Results and the Methods sections. We have decided not to add another supplementary figure since we believe that the current wording should be sufficient in making this point.

The following changes were made in the Results section (changes underlined):

However, another source of error that may vary with set size is that participants may mistakenly recall the location of an object other than the one they attempted to recall. For example, participants may suffer a memory confusion among the different cats and mistakenly place one cat (the Persian cat) in the location of another from the same set (the Siamese cat). We therefore designed the study in a manner that allowed us to dissociate these errors, which we term swap errors from placement errors that more directly reflect degree of spatial accuracy in recall (see Methods).

The following changes were made in the Methods section (changes underlined):

For the spatial-memory task, we define two types of placement errors: swap errors and accuracy errors. Swap errors are errors in which a participant mistakenly positioned an image near the location of another image belonging to the same set. If an item was positioned closer to another same-set item's position, it was considered swapped. Accuracy errors, measured in pixels, reflect the distance between item placement and its correct position. Our design allowed us to disentangle swap errors and accuracy errors (i.e., correct item locations within a set were far apart, as described above). For each item placement in the pre- and post-nap tests, we first considered whether there was a swap error. Only non-swapped items were considered for the accuracy error analyses. The rationale behind detecting and removing swap errors from accuracy estimates is that they inflate errors in a manner that does not purely reflect spatial-memory.

A caveat of our method of detecting swap errors is that pure guesses may also be counted as swap errors if they happen to be closer to another image location than to the correct one. We were aware of this limitation, but preferred to overestimate rather than under-estimate swap errors to avoid their contamination of accuracy errors. However, our definition of swap errors could create an unfair advantage in error rate for bigger sets. For example, a large error due to guessing (e.g., 800 pixels) would more likely have been closer to some other item in a six-item set and be counted as a swap error relative to a two-item set. The accuracy error for this item would therefore not have been considered for the larger set. The same error for an item in a single-item set would not have another item to swap with and the accuracy error would have been considered, inflating errors for smaller sets. To compensate for this bias, swap errors were calculated by taking into consideration six locations, even for sets of one or two items. For example, a swap error in a one-item set would have taken place if the participant mistakenly positioned the single item closer to any of five other locations on the grid, despite the fact that these locations were never linked to any item.

These alternative locations (five alternatives for the one-item sets and four for the two-item sets) were at least 400 pixels apart from each other and from the correct locations of the items in the set. All swapped items for all set sizes were removed from the accuracy error analyses. Note that larger sets would still have more identified swap errors, because this value encompassed both large guessing-related errors (which are presumably identical for items of different set sizes) and errors due to item swapping (which should be higher for larger sets). The average number of swap errors, which is equivalent to the number of items removed, is presented in the Results section for the two- and six-item sets. The average number of swap errors for one-item sets was 0.1 ± 0.01 before sleep and 0.09 ± 0.02 after sleep.

Signed,
Michael Scullin

Reviewer #3 (Remarks to the Author):

COMMSBIO-20-1738-T

Multiple memories can be simultaneously reactivated during sleep as effectively as a single memory

Schechtman, E., Antony, J. W., Lampe, A., Wilson, B. J., Norman, K. A. & Paller, K. A.

This study examined whether the effect of cued reactivation during sleep on memory depends on the amount of information being reactivated via cues during sleep. It aimed at elucidating whether cued reactivation of multiple memories entails costs. The study further examined the contribution of sleep-related brain oscillations in processing reactivated memories. One group of 31 participants encoded 90 picture-location-sound associations, which were grouped in 6-, two- or one-pictures-sets of different themes (e.g. 6 different cats, two different parts of airplanes). Each picture was associated with a specific theme sound (e.g. all cats – meow). After learning the positions of the pictures to criterion, participants slept and were presented with half of the sounds and one new sound during non-REM sleep (mostly SWS, very few S2), in order to reactivate single or multiple memories. Afterwards, correct placement of the pictures was tested. Results showed that set size did not modulate the success of reactivation for placement memory, but reactivated sets were overall recalled better than not reactivated sets. Cueing affected the delta-theta and the sigma frequency bands. Larger sets had a stronger modulatory impact on these bands with higher spindle probability associated with larger set size. But delta-theta and sigma frequency bands and set size were not associated with memory performance.

Overall, these findings are novel and highly interesting to the field of cued reactivation of memories during sleep in showing that the success of cued reactivation is not attenuated by the reactivation of multiple items compared to a single item and that set size during learning is associated with oscillations in the sigma and delta frequencies, while these frequencies are not related to memory performance.

The methods are sound, statistics are elaborated and convincing, the manuscript is well-written and results are discussed appropriately. I have only a few comments that should be addressed.

Major points:

1. My first concern relates to the prediction of the cueing effect by physiological measures. In the discussion section, the authors mention that the cumulative benefit over items for the behavioral analysis is modulated by set size (see Discussion section, p. 19, line 422). Please add this analysis to the results section. Beyond that, do delta-theta power, sigma power and spindle probability predict this cumulative measure of cueing benefit (there is no such effect when predicting the averaged cueing benefit, see Results section, p. 14, lines 305 - 309). Please, discuss the (lacking) effect of physiological measures on (cumulative) cueing benefits.

We thank the reviewer for this valuable critique.

Behaviorally, taking into account the cumulative benefits across sets, we now find a marginal interaction effect ($p < 0.06$).

The following paragraph was added to the Results section (changes underlined):

Our results show that the cuing-related benefit of a single item does not depend on the size of the set to which it belongs. To complement this analysis, we considered the cumulative benefit for sets of different sizes by computing the sum of all benefits aggregated over the different items within a set. Like the per-item analysis, the per-set analysis also revealed a significant effect of cuing ($F(1,30)=9.83$, $p<0.01$, $\eta^2=0.25$) and no main effect of set size ($F(2,60)=0.35$, $p=0.71$, $\eta^2=0.01$). However, unlike the per-item analysis, this analysis revealed a marginal interaction ($F(2,60)=3.02$, $p=0.056$, $\eta^2=0.09$), indicating a trend for larger cumulative benefits for larger sets relative to smaller ones. Given similar benefits for single items within a set, at the level of whole sets it would be reasonable for a set with many items to benefit more than a set with few items.

The following paragraph was corrected in the Discussion section (changes underlined):

Our results showed a seemingly discrepant pattern: the behavioral finding of set-size-independent memory benefits on the one side and the physiological finding of set-size-dependent increases in delta-theta and sigma power on the other. However, the set-size-independent memory effect held only for the average benefit per item (averaged across one, two, or six items). When considering the cumulative benefit combined over all items in a set, there was a marginally larger cumulative benefit for larger sets. Delta-theta and sigma power may therefore represent the cumulative benefit for the set associated with the presented cue. Alternatively, the physiological measures may represent the process by which memories are retrieved and modified—in which case delta-theta and sigma power may reflect the extent of previous knowledge made available for consolidation. Either way, the physiological and behavioral findings can be reconciled based on these considerations and thus are not at odds with each other.

Correlating the cumulative benefits with our physiological data proved somewhat challenging. At face value, when conducting this analysis, it seems as though the modulation coefficients (i.e., the betas estimated by the mixed linear model) of both the sigma power and the sigma power x set size interaction are significant ($t(5986)=-2.14$, $p<0.04$, modulation coefficient = -0.05, 95% CI: -0.1– -0.004 for main effect; $t(5986)=3.06$, $p<0.003$, modulation coefficient = -0.02, 95% CI: 0.01–0.04 for interaction). However, we are concerned that these results may be an inherent artifact of our analysis. Trivially, the range of the cumulative behavioral benefits is different for sets of six, two and one items (*range*: -410 to 818 pixels for 6-item sets, -298 to 314 pixels for 2-item sets, -139 to 168 pixels for 1-item sets; *standard deviation*: 175 pixels for 6-item sets, 81 pixels for 2-item sets, 44 pixels for 1-item sets). Since the potential for correlation depends on the variability of the data and since this variability is larger for datasets with wider ranges, plugging this data in a single model may erroneously exaggerate the influence of sigma power on the six-item set, which is indeed what our results seem to show. We are not aware of a method to fully account for this issue and still take into account all the data in a manner that allows for estimating differences in correlation between set sizes. Importantly, the correlation between sigma power and cumulative benefits limited to the six item sets is not significant, again suggesting that our method may artificially inflate these (possibly nonexistent) effects. We have therefore decided not to include these results in the manuscript, to avoid misleading our readers.

We have attached here a figure displaying these results. (a) 3d histograms showing the co-distribution of cumulative change in error over sleep and sigma power modulation for

different set size. (b) Regression lines based on coefficients derived from a linear mixed model. The sigma power modulation and the set-size x sigma power interaction effects are significant. The model used for this analysis is $CumulativeMemoryBenefit \sim 1 + Modulation*SetSize + (Modulation|SubjectID)$.

Analyses for delta-theta power and spindle probability are equally sensitive to this caveat, but they were not significant to begin with ($p > 0.27$).

2. My second point relates to the semantic relatedness of the stimuli within a set. It seems that different cats are related more closely than different parts of an airplane (example see Methods section, Materials, p. 22, line 472). Was semantic relatedness equally distributed among set sizes and among cued and not cued sets (i.e. among two- and six-item sets)?

In our design, we aimed to choose items that maximized semantically relatedness within set. We did not test semantic relatedness explicitly, but we acknowledge that it may have varied

between different sets. However, importantly, these differences should not have biased our results in any way, although they may have introduced noise. The selection of which sets were cued and which were not was randomly made by our algorithm for each participant, so it is fair to assume that the level of relatedness was balanced. To avoid differences in levels of relatedness between sets of two and six items, we created a pool of six-item sets from which both six- and two-item sets were randomly selected. For the two-item sets, two of the six available items were randomly chosen. In summary, we believe that any variability in relatedness may have introduced noise and decreased power, but it is very unlikely that it biased our results in any way.

We now address this issue in our Discussion (changes underlined):

Another related point concerns variability in relatedness between different sets that were used in our study (e.g., it could be that the “Cat” set was more interrelated than the “Airplane” set). Importantly, these differences should not have biased our results in any way, although they may have introduced noise and therefore decreased power. The selection of which sets were cued and which were not was accomplished with an algorithm to effectively balance between-set memory differences.

3. Related to my previous point, do the authors expect any effects of semantic relatedness on their results? E.g., cued reactivation might be more effective for highly related sets if cued reactivation occurs in a generalized context manner, while it might be independent of semantic relatedness if cued reactivation occurs for individual memories. Please consider discussing this possibility and if suitable, include results.

This is an important question that we hope to address in a future study. We hope our paper will encourage others to explore this path as well. We have now addressed this point in our Discussion (changes underlined):

Our results open the door to further investigation of the importance of relatedness to sleep-related memory consolidation. Future studies should attempt to manipulate item relatedness to reveal the boundary conditions of simultaneous reactivation during sleep.

Minor points:

1. As far as I understand, the ANOVA testing the effect of set size and cueing on change in swap errors included the main effect of set size. For the sake of completeness, please report results of this main effect (see Results section, p. 7, lines 141 – 146). The same accounts for the ANOVA testing the effect of the same factors on change in spatial errors controlled for pre-sleep errors (see Results section, p 8, line 170-172) and for caption of Figure 2 (see Results section, p. 40, “The lower panel shows...”). Also, consider reporting the effect of these factors on pre-sleep swap error, since pre-sleep errors were corrected for swap errors and differ depending on set size.

We thank the reviewer for these constructive suggestions. We have made the following changes to the manuscript (changes underlined):

Reporting difference in pre-sleep swapping errors across conditions and adding missing main effect:

Focusing first only on sets with two or six items, we considered the effects of set size and cuing during sleep on swap errors. The average number of swap errors for an item in a six-item set was 0.385 ± 0.02 before sleep and 0.391 ± 0.02 after sleep (change = +0.005). The average number of swap errors for an item in a two-item set was 0.175 ± 0.02 before sleep and 0.178 ± 0.02 after sleep (change = +0.003). Comparing the effects of set size and future-cuing status on pre-sleep swap errors using a repeated-measures ANOVA revealed a significant effect of set size, with larger sets having more swap errors ($F(1,30)=58.9$, $p<0.001$, $\eta^2=0.66$). Importantly, there were no pre-sleep differences between cued and non-cued sets ($F(1,30)=0.006$, $p=0.94$, $\eta^2=0.0002$), nor was there a significant interaction ($F(1,30)=0.08$, $p=0.78$, $\eta^2=0.003$). We next turned to changes to memory between the pre- and post-sleep tests. Using a repeated-measures ANOVA, we found no main effect of cuing on the change in swap errors ($F(1,30)=0.43$, $p=0.52$, $\eta^2=0.014$), no main effect of set size ($F(1,30)=0.03$, $p=0.87$, $\eta^2=0.0009$), and no interaction between cuing and set size ($F(1,30)=0.24$, $p=0.63$, $\eta^2=0.008$).

Adding missing main effect for the analysis controlling for pre-sleep errors:

This analysis produced similar results, with a significant effect for cuing ($F(1,30)=14.75$, $p<0.001$, $\eta^2=0.33$), no significant effect of set size ($F(1,30)=2.16$, $p=0.12$, $\eta^2=0.07$), and no significant interaction between set size and cuing status ($F(2,60)=0.29$, $p=0.75$, $\eta^2=0.01$; Supplementary Figure 2a).

Adding missing main effect to caption of Figure 2:

Negative values represent higher errors after sleep. There was a significant effect of cuing status but no significant effect of set size and no cuing by set size interaction. Error bars signify standard error of the mean (between-subjects).

2. Standard errors should be added to the averaged spatial-memory errors for cued and non-cued pictures (see Results section, p. 8, line 164).

This was now added (changes underlined):

On average, spatial-memory error after sleep decreased by 2.98 ± 1.87 pixels for cued sets and increased by 4.76 ± 2.39 pixels for non-cued sets.

3. The reference to Figure 4c) is missing in the description of the results of the linear mixed model (see Results section, p. 13, lines 279 - 284).

This has now been corrected. Thank you for noticing this oversight (changes underlined):

Results showed that power for both the delta-theta and the sigma clusters was linearly modulated by the number of items previously associated with the sound (delta-theta: $t(6889)=2.52$, $p<0.02$, modulation coefficient = 1.97, 95% CI: 0.44–3.51; sigma: $t(6889)=2.08$, $p<0.04$, modulation coefficient = 1.99, 95% CI: 0.11–3.87; Figure 4c).

4. It is not clear to me, what exactly is meant by the random effect of sound identification in the description of the linear mixed model estimating the effect of set size on spectral power (see Methods section, p. 30, line 654). Please clarify.

We thank the reviewer for pointing this possible source of confusion. Participants were exposed to multiple sounds, each presented several times. The individual sound-related modulation values were therefore nested not only within participants, but within each sound identifier.

This is now briefly explained in the manuscript, in addition to a more specific description of the model (changes underlined):

To estimate the effect of the number of items associated with each sound on spectral power, we used linear mixed models with the number of items (i.e., zero, one, two and six) as a fixed effect. Two random effects were accounted for: participant (i.e., to account for random differences between participants) and sound identifications (i.e., to account for random differences between sound-specific EEG responses within participant). The models we used is therefore the following:

$$(1) \text{ Modulation} \sim 1 + \text{SetSize} + (1 + \text{SetSize}|\text{SubjectID}) + (1|\text{SoundID})$$

where Modulation is the change in sigma or delta-theta power. This nested model produced an estimate of the linear coefficient for the delta-theta and sigma clusters (Figure 4c).

5. Please, specify the linear mixed model to predict cueing benefit (see Results section, p. 14, lines 304 – 310).

The following paragraph was added:

Finally, another mixed linear model was used to reveal the effects of the physiological modulations on both per-item and per-set cueing benefits. For this, the following model was used:

$$(1) \text{ MemoryBenefit} \sim 1 + \text{Modulation} * \text{SetSize} + (\text{Modulation}|\text{SubjectID})$$

6. Figure 2: please, add * to indicate significant results.

The appropriate changes were made (see below).

REVIEWERS' COMMENTS:

Reviewer #1 (Remarks to the Author):

The authors addressed all my comments. I found the revision to be thoughtful and professional, and I expect this work to be impactful.

Reviewer #3 (Remarks to the Author):

COMMSBIO-20-1738A

Multiple memories can be simultaneously reactivated during sleep as effectively as a single memory

Schechtman, E., Antony, J. W., Lampe, A., Wilson, B. J., Norman, K. A. & Paller, K. A.

All comments have been addressed appropriately by the authors and I can recommend the manuscript for publication in Communications Biology.